# INFORMATION THEORETIC MODEL PREDICTIVE Q-LEARNING

## ABSTRACT

Model-free reinforcement learning (RL) algorithms work well in sequential decision-making problems when experience can be collected cheaply and model-based RL is effective when system dynamics can be modeled accurately. However, both of these assumptions can be violated in real world problems such as robotics, where querying the system can be prohibitively expensive and real-world dynamics can be difficult to model accurately. Although sim-to-real approaches such as domain randomization attempt to mitigate the effects of biased simulation, they can still suffer from optimization challenges such as local minima and hand-designed distributions for randomization, making it difficult to learn an accurate global value function or policy that directly transfers to the real world. In contrast to RL, model predictive control (MPC) algorithms use a simulator to optimize a simple policy class online, constructing a closed-loop controller that can effectively contend with real-world dynamics. MPC performance is usually limited by factors such as model bias and the limited horizon of optimization. In this work, we present a novel theoretical connection between information theoretic MPC and entropy regularized RL and develop a Q-learning algorithm that can leverage biased models. We validate the proposed algorithm on sim-to-sim control tasks to demonstrate the improvements over optimal control and reinforcement learning from scratch. Our approach paves the way for deploying reinforcement learning algorithms on real-robots in a systematic manner.

## 1 INTRODUCTION

Deep reinforcement learning algorithms have recently generated great interest due to their successful application to a range of difficult problems including Computer Go (Silver et al., 2016) and high-dimensional control tasks such as humanoid locomotion (Lillicrap et al., 2015; Schulman et al., 2015). While these methods are extremely general and can learn policies and value functions for complex tasks directly from data, they can also be sample inefficient, and partially-optimized solutions can be arbitrarily poor. These challenges severely restrict RL's applicability to real systems such as robots due to data collection challenges and safety concerns.

One straightforward way to mitigate these issues is to learn a policy or value function entirely in a high-fidelity simulator (Shah et al., 2017; Todorov et al., 2012) and then deploy the optimized policy on the real system. However, this approach can fail due to model bias, external disturbances, the subtle differences between the real robot's hardware and poorly modeled phenomena such as friction and contact dynamics. Sim-to-real transfer approaches based on domain randomization (Sadeghi & Levine, 2016; Tobin et al., 2017) and model ensembles (Kurutach et al., 2018; Shyam et al., 2019) aim to make the policy robust by training it to be invariant to varying dynamics. However, learning a globally consistent value function or policy is hard due to optimization issues such as local optima and covariate shift between the exploration policy used for learning the model and the actual control policy executed on the task (Ross & Bagnell, 2012).

Model predictive control (MPC) is a widely used method for generating feedback controllers that repeatedly re-optimizes a finite horizon sequence of controls using an approximate dynamics model that predicts the effect of these controls on the system. The first control in the optimized sequence is executed on the real system and the optimization is performed again from the resulting next state. However, the performance of MPC can suffer due to approximate or simplified models and lim-

ited lookahead. Therefore the parameters of MPC, including the model and horizon $H$ should be carefully tuned to obtain good performance. While using a longer horizon is generally preferred, real-time requirements may limit the amount of lookahead and a biased model can result in compounding model errors.

In this work, we present an approach to RL that leverages the complementary properties of model-free reinforcement learning and model-based optimal control. Our proposed method views MPC as a way to simultaneously approximate and optimize a local Q function via simulation, and Q learning as a way to improve MPC using real-world data. We focus on the paradigm of entropy regularized reinforcement learning where the aim is to learn a stochastic policy that minimizes the cost-to-go as well as KL divergence with respect to a prior policy. This approach enables faster convergence by mitigating the over-commitment issue in the early stages of Q-learning and better exploration (Fox et al., 2015). We discuss how this formulation of reinforcement learning has deep connections to information theoretic stochastic optimal control where the objective is to find control inputs that minimize the cost while staying close to the passive dynamics of the system (Theodorou & Todorov, 2012). This helps in both injecting domain knowledge into the controller as well as mitigating issues caused by over optimizing the biased estimate of the current cost due to model error and the limited horizon of optimization. We explore this connection in depth and derive an infinite horizon information theoretic model predictive control algorithm based on Williams et al. (2017). We test our approach called Model Predictive Q Learning (MPQ) on simulated continuous control tasks and compare it against information theoretic MPC and soft Q-Learning (Haarnoja et al., 2017), where we demonstrate faster learning with fewer system interactions and better performance as compared to MPC and soft Q-Learning even in the presence of sparse rewards. The learned Q function allows us to truncate the MPC planning horizon which provides additional computational benefits. Finally, we also compare MPQ versus domain randomization(DR) on sim-to-sim tasks. We conclude that DR approaches can be sensitive to the hand-designed distributions for randomizing parameters which causes the learned Q function to be biased and suboptimal on the true system's parameters, whereas learning from data generated on true system is able to overcome biases and adapt to the real dynamics.

## 2 RELATED WORK

Model predictive control has a rich history in robotics, ranging from control of mobile robots such as quadrotors (Desaraju & Michael, 2016) and aggressive autonomous vehicles (Wagener et al., 2019; Williams et al., 2017) to generating complex behaviors for high-dimensional systems such as contact-rich manipulation (Fu et al., 2016; Kumar et al., 2014) and humanoid locomotion (Erez et al., 2013). The success of MPC can largely be attributed to online policy optimization which helps mitigate model bias. The information theoretic view of MPC aims to find a policy at every timestep that minimizes the cost over a finite horizon as well as the KL-divergence with respect to a prior policy usually specified by the system's passive dynamics (Theodorou & Todorov, 2012; Williams et al., 2017). This helps maintain exploratory behavior and avoid over-commitment to the current estimate of the cost function, which is biased due to modeling errors and a finite horizon. Sampling-based MPC algorithms (Wagener et al., 2019; Williams et al., 2017) are also highly parallelizable enabling GPU implementations that aid with real-time control. However, efficient MPC implementations still require careful system identification and extensive amounts of manual tuning.

Deep RL methods are extremely general and can optimize neural network policies from raw sensory inputs with little knowledge of the system dynamics. Both value-based and policy-based approaches (Schulman et al., 2015) have demonstrated excellent performance on complex control problems. These approaches, however, fall short on several accounts when applying them to a real robotic system. First, they have high sample complexity, potentially requiring millions of interactions with the environment. This can be very expensive on a real robot, not least because the initial performance of the policy can be arbitrarily bad. Using random exploration methods such a $\epsilon$-greedy can further aggravate this problem. Second, a value function or policy learned entirely in simulation inherits the biases of the simulator. Even if a perfect simulation is available, learning a globally consistent value function or policy is an extremely hard task as noted in (Silver et al., 2016; Zhong et al., 2013). This can be attributed to local optima when using neural network representations or the inherent biases in the Q learning update rules (Fox et al., 2015; Van Hasselt et al., 2016). In fact, it can be difficult to explain why Q-learning algorithms work or fail (Schulman et al., 2017).

Domain randomization aims to make policies learned in simulation more robust by randomizing simulation parameters during training with the aim of making the policies invariant to potential parameter error (Peng et al., 2018; Sadeghi & Levine, 2016; Tobin et al., 2017). However, these policies are not *adaptive* to unmodelled effects, i.e they take into account only aleoteric and not epistemic uncertainty. Also, such approaches are highly sensitive to hand-designed distributions used for randomizing simulation parameters and can be highly suboptimal on the real-systems parameters, for example, if a very large range of simulation parameters is used. Model-based approaches aim to use real data to improve the model of the system and then perform reinforcement learning or optimal control using the new model or ensemble of models (Kurutach et al., 2018; Ross & Bagnell, 2012; Shyam et al., 2019). Although learning accurate models is a promising avenue, we argue that learning a globally consistent model is an extremely hard problem and instead we should learn a policy that can rapidly adapt to experienced real-world dynamics.

The use of entropy regularization has been explored in RL and Inverse RL for its better sample efficiency and exploration properties (Fox et al., 2015; Haarnoja et al., 2017; 2018; Schulman et al., 2017; Ziebart et al., 2008). This framework allows incorporating prior knowledge into the problem and learning multi-modal policies that can generalize across different tasks. Fox et al. (2015) analyze the theoretical properties of the update rule derived using mutual information minimization and show that this framework can overcome the over-estimation issue inherent in the vanilla Q-learning update. In the past, Todorov (2009) have shown that using KL-divergence can convert the optimal control problem into one that is linearly solvable.

Infinite horizon MPC aims to learn a terminal cost function that can add global information to the finite horizon optimization. Rosolia & Borrelli (2017) learn a terminal cost as a control Lyapunov function and a safety set for the terminal state. These quantites are calculated using all previously visited states and they assume the presence of a controller that can deterministically drive the any state to the goal. Tamar et al. (2017) learns a cost shaping to make a short horizon MPC mimic the actions produced by long horizon MPC offline. However, since their approach is to mimic a longer horizon MPC, the performance of the learner is fundamentally limited by the the performance of the longer horizon MPC. On the contrary, learning an optimal value function as the terminal cost can potentially lead to close to optimal performance.

Using local optimization is an effective way of improving an imperfect value function as noted in RL literature by Anthony et al. (2017); Lowrey et al. (2018); Silver et al. (2016; 2017); Sun et al. (2018). However, these approaches assume that a perfect model of the system is available. In order to make the policy work on the real system, we argue that it is essential to learn a value function form real data and utilize local optimization to stabilize learning.

## 3 PRELIMINARIES

### 3.1 REINFORCEMENT LEARNING WITH ENTROPY REGULARIZATION

A Markov Decision Process (MDP) is defined by tuple $(\mathcal{S}, \mathcal{A}, c, \mathcal{P}, \gamma)$ where $\mathcal{S}$ is the state space, $\mathcal{A}$ is the action space, $c$ is a one step cost function, $\mathcal{P}$ is the space of transition functions and $\gamma$ is a discount factor. Let $P \in \mathcal{P}$ be a particular transition function. A closed-loop policy $\pi(a|s)$ is a distribution over actions given state. Given a policy $\pi$ and a *prior* policy $\bar{\pi}$, the KL divergence between them at a state is given by $KL\left(\pi(a|s)||\bar{\pi}(a|s)\right) = \mathbb{E}_\pi\left[\log\left(\pi(a|s)/\bar{\pi}(a|s)\right)\right]$. Entropy-regularized RL (Fox et al., 2015) aims to optimize the following objective

$$\pi^* = \arg\min_\pi \mathbb{E}_{\pi,P}\left[\sum_{t=1}^\infty \gamma^{t-1}\left(c(s_t, a_t) + \lambda KL\left(\pi_t||\bar{\pi}_t\right)\right)\right] \ \forall \ s_0 \ \in \ \mathcal{S} \quad (1)$$

where $\pi_t$ and $\bar{\pi}_t$ are shorthand for $\pi(a_t|s_t)$ and $\bar{\pi}(a_t|s_t)$ respectively, $\lambda$ is the temperature parameter that penalizes deviation of $\pi$ from $\bar{\pi}$. For a policy $\pi$, we can define the *soft* value and action-value functions

$$V^\pi(s) = \mathbb{E}_{\pi,P}\left[\sum_{t=1}^\infty \gamma^{t-1}\left(c(s_t, a_t) + \lambda KL\left(\pi_t||\bar{\pi}_t\right)\right)|s_0 = s\right]$$
$$Q^\pi(s,a) = c(s,a) + \gamma\mathbb{E}_{s'\sim P(s'|s,a)}\left[V^\pi(s')\right] \quad (2)$$

Given a horizon of $H$ timesteps, we can use above definitions to write the value functions as

$$V^\pi(s) = \mathbb{E}_{\pi,P} \left[ \sum_{t=1}^{H-1} \gamma^{t-1}(c(s_t, a_t) + \lambda KL\left(\pi_t || \bar{\pi}_t\right)) + \gamma^{H-1} V^\pi(s_H) | s_1 = s \right]$$

$$Q^\pi(s, a) = c(s, a) + \mathbb{E}_{\pi,P} \left[ \sum_{t=2}^{H-1} \gamma^{t-1}(c(s_t, a_t) + \lambda KL\left(\pi_t || \bar{\pi}_t\right)) \right.$$

$$\left. + \gamma^{H-1}(\lambda KL\left(\pi_H || \bar{\pi}_H\right) + Q(s_H, a_H)) | s_1 = s, a_1 = a \right] \quad (3)$$

It is straightforward to verify that $V^\pi(s) = \mathbb{E}_{a \sim \pi} \left[ \log(\pi(a|s)/\bar{\pi}(a|s)) + Q(s, a) \right]$. The objective in Eq. (1) can equivalently be written as

$$\pi^* = \arg\min_\pi V^\pi(s) \ \forall \ s \ \in \ \mathcal{S} \quad (4)$$

This optimization can be performed either by policy gradient methods that aim to find the optimal policy $\pi \in \Pi$ via stochastic gradient descent (Schulman et al., 2017; Williams, 1992) or value based methods that try to iteratively approximate the value function of the optimal policy. In either case, the output of solving the above optimization is a global closed-loop control policy $\pi^*(a|s)$.

## 3.2 Information Theoretic MPC

Solving the above optimization can be prohibitively expensive and hard to accomplish online. In contrast to RL, MPC performs online optimization of a simple policy class with a truncated horizon. This process effectively creates a closed-loop controller. In order to do so, MPC algorithms such as MPPI (Williams et al., 2017) use an approximate dynamics model $\hat{P}$, which can be deterministic. This is the case when using a simulator such as MuJoCo (Todorov et al., 2012) as the dynamics model. At timestep $t$, starting from the current state $s_t$, an open loop sequence of actions $A = (a_t, a_{t+1}, \dots a_{t+H})$ is sampled from the control distribution denoted by $\pi(A)$. The objective is to find an optimal sequence of actions to solve

$$A^* = \arg\min_A \mathbb{E}_{\pi(A)} \left[ \sum_{l=t}^{t+H-1} \gamma^{l-t} c(s_l, a_l) + \lambda KL\left(\pi_l || \bar{\pi}_l\right) + \gamma^{H-1}(c_f(s_{t+H}, a_{t+H}) + \lambda KL\left(\pi_{t+H} || \bar{\pi}_{t+H}\right)) \right]$$
$$(5)$$

where $c_f(s_{t+H}, a_{t+H})$ is a terminal cost function and $\bar{\pi}(A)$ is the passive dynamics of the system, i.e the distribution over actions produced when the control input is zero. The first action in the sequence is then executed on the system and the optimization is performed again from the resulting next state. The re-optimization and entropy regularization helps in mitigating model-bias and inaccuracies with optimization by avoiding overcommitment to the current estimate of the cost. A shortcoming of the MPC procedure is the finite horizon. This is especially pronounced in tasks with sparse rewards where a short horizon can make the agent myopic to future rewards. To mitigate this, an approach known as infinite horizon MPC sets the terminal cost $c_f$ as a value function that adds global information to the problem.

In the next section, we build our approach by focussing on the MPPI algorithm and its relationship with entropy regularized reinforcement learning. Specifically, we use the definitions of the soft value functions from Eq. (2) to derive an optimal Boltzmann distribution for $H$-step actions that optimally solves the infinite horizon control problem. This helps us derive the MPPI update rule from Williams et al. (2017) for the infinite horizon case, which then leads to our Model Predictive Q Learning (MPQ) algorithm, which utilizes a predictive model for Q updates and stochastic optimal control as the policy. In the case where $H = 1$, the algorithm is equivalent to soft Q learning (Haarnoja et al., 2018) or G-learning (Fox et al., 2015).

## 4 Approach

### 4.1 Optimal H-step Boltzmann Distribution

Let $\pi(A)$ and $\bar{\pi}(A)$ be the joint control distribution and prior over $H$-horizon open-loop actions. The distributions are assumed to be independent over timesteps, i.e $\pi(a_1 \dots a_H) = \prod_{t=1}^{H} \pi_t$, where

$\pi_t$ is shorthand for $\pi(a_t)$. Since $\hat{P}$ is deterministic, the following equations hold

$$V^\pi(s) = \mathbb{E}_\pi\left[\lambda\log(\pi/\bar{\pi}) + Q^\pi(s,a)\right] \qquad\qquad Q^\pi(s,a) = c(s,a) + \gamma V^\pi(s') \qquad (6)$$

For clarity, we consider $\gamma = 1$. Substituting from Eq. (3) for $Q^\pi(s,a)$

$$V^\pi(s) = \mathbb{E}_{\pi_1\dots\pi_H}\left[\sum_{t=1}^{H-1} c(s_t,a_t) + \lambda\sum_{t=1}^{H}\log(\pi_t/\bar{\pi}_t) + Q^\pi(s_H,a_H)\right]$$

$$= \mathbb{E}_{\pi_1\dots\pi_H}\left[\sum_{t=1}^{H-1} c(s_t,a_t) + \lambda\log\prod_{t=1}^{H}(\pi_t/\bar{\pi}_t) + Q^\pi(s_H,a_H)\right]$$

$$= \mathbb{E}_\pi\left[\sum_{t=1}^{H-1} c(s_t,a_t) + \lambda\log(\pi/\bar{\pi}) + Q^\pi(s_H,a_H)\right] \qquad (7)$$

Consider the following distribution over H-horizon

$$\pi = \frac{1}{\eta}\exp\left(\frac{-1}{\lambda}\left(\sum_{t=1}^{H-1} c(s_t,a_t) + Q^\pi(s_H,a_H)\right)\right)\bar{\pi}(a_1\dots a_H) \qquad (8)$$

where $\eta$ is a normalization constant given by

$$\eta = \mathbb{E}_{\bar{\pi}(a_1\dots a_H)}\left[\exp\left(\frac{-1}{\lambda}\left(\sum_{t=1}^{H-1} c(s_t,a_t) + Q^\pi(s_H,a_H)\right)\right)\right] \qquad (9)$$

We show that this is the optimal control distribution as $\nabla V^\pi(s) = 0$. Substituting Eq. (8) in (7)

$$V^\pi(s) = \mathbb{E}_\pi\left[\sum_{t=1}^{H-1} c(s_t,a_t) - \lambda\log\eta - \sum_{t=1}^{H-1} c(s_t,a_t) - Q^\pi(s_H,a_H) + Q^\pi(s_H,a_H)\right]$$
$$V^\pi(s) = \mathbb{E}_\pi\left[-\lambda\log\eta\right]$$

Since $\eta$ is a constant, we have $V^\pi(s) = -\lambda\log\eta$. Hence for $\pi$ in Eq. (8), the soft value function is a constant with gradient zero given by

$$V^{\pi^*}(s) = -\lambda\log\mathbb{E}_{\bar{\pi}(a_1\dots a_H)}\left[\exp\left(\frac{-1}{\lambda}\left(\sum_{t=1}^{H-1} c(s_t,a_t) + Q^\pi(s_H,a_H)\right)\right)\right] \qquad (10)$$

which is often referred to in optimal control literature as the "free energy" of the system (Theodorou & Todorov, 2012; Williams et al., 2017). For H=1, Eq. (10) takes the form of the soft value function from Fox et al. (2015) and Haarnoja et al. (2018).

## 4.2 Infinite Horizon MPPI Update Rule

Similar to Williams et al. (2017), we derive the MPPI update rule for online policy optimization. Since sampling actions from the optimal control distribution in Eq. (8) is intractable, we consider control policies $\pi(A) \in \Pi$ which are easy to sample from. We then optimize for a vector of $H$ control inputs $U$, such that the resulting action distribution minimizes the KL divergence with the optimal policy

$$U^* = \underset{\pi(A)}{\arg\min}\, KL\left(\pi^*(A)||\pi(A)\right) \qquad (11)$$

The objective can be expanded out as

$$KL\left(\pi^*(A)||\pi(A)\right) = \int_A \pi^*(A)\log\frac{\pi^*(A)}{\pi(A)}\mathrm{d}A = \int_A \pi^*(A)\log\frac{\pi^*(A)}{\bar{\pi}(A)}\frac{\bar{\pi}(A)}{\pi(A)}\mathrm{d}A$$

$$= \int_A \pi^*(A)\log\frac{\pi^*(A)}{\bar{\pi}(A)}dA - \int_A \pi^*(A)\log\frac{\bar{\pi}(A)}{\pi(A)}\,\mathrm{d}A \qquad (12)$$

Since the first term does not depend on the control input, we can remove it from the optimization

$$U^* = \underset{\pi(A)}{\arg\max} \int_A \pi^*(A) \log \frac{\bar{\pi}(A)}{\pi(A)} \, \mathrm{d}A \tag{13}$$

Consider $\Pi$ to be independent multivariate Gaussians over sequence of the $H$ controls with constant covariance $\Sigma$ at each timestep. We can write the control distribution and prior as follows

$$\pi(A) = \frac{1}{Z} \prod_{t=1}^{H} \exp\left(-\frac{1}{2}(u_t - a_t)^T \Sigma^{-1}(u_t - a_t)\right) \quad \bar{\pi}(A) = \frac{1}{Z} \prod_{t=1}^{H} \exp\left(-\frac{1}{2}a_t^T \Sigma^{-1} a_t\right) \tag{14}$$

where $u_t$ and $a_t$ are the control inputs and actions at timestep $t$ and $Z$ is the normalization constant. Here the prior corresponds to the passive dynamics of the system (Theodorou & Todorov, 2012; Williams et al., 2017), although other choices of prior are possible. Substituting in Eq. (13) we get

$$U^* = \underset{\pi(A)}{\arg\max} \int \pi^*(A) \left(\sum_{t=1}^{H} -\frac{1}{2}u_t^T \Sigma^{-1} u_t + u_t^T \Sigma^{-1} a_t\right) \, \mathrm{d}A \tag{15}$$

The objective can be simplified to the following by integrating out the probability in the first term

$$\sum_{t=1}^{H} -\frac{1}{2}u_t^T \Sigma^{-1} u_t + u_t^T \int \pi^*(A) \Sigma^{-1} a_t \, \mathrm{d}A \tag{16}$$

Since this is a concave function with respect to every $u_t$, we can find the maximum by setting its gradient with respect to $u_t$ to zero to solve for optimal $u_t^*$

$$u_t^* = \int \pi^*(A) a_t \mathrm{d}A = \int \pi(A) \frac{\pi^*(A)}{\bar{\pi}(A)} \frac{\bar{\pi}(A)}{\pi(A)} a_t \mathrm{d}A = \mathbb{E}_{\pi(A)}\left[\frac{\pi^*(A)}{\bar{\pi}(A)} \frac{\bar{\pi}(A)}{\pi(A)} a_t\right] = \mathbb{E}_{\pi(A)}\left[w(A) a_t\right] \tag{17}$$

where the second equality comes from importance sampling to convert the optimal controls into an expectation over the control distribution instead of the optimal distribution which is impossible to sample from. The importance weight $w(A)$ can be written as follows (substituting $\pi^*$ from Eq. (8))

$$w(A) = \frac{1}{\eta} \mathbb{E}_{\pi(A)}\left[\exp\left(\frac{-1}{\lambda}\left(\sum_{t=1}^{H-1} c(s_t, a_t) + Q^{\pi^*}(s_H, a_H)\right)\right) \frac{\bar{\pi}(A)}{\pi(A)}\right] \tag{18}$$

Making change of variables $u_t + \epsilon_t = a_t$ for noise sequence $\mathcal{E} = (\epsilon_1 \ldots \epsilon_H)$ sampled from independant Gaussians with zero mean and covariance $\Sigma$ we get

$$w(\mathcal{E}) = \frac{1}{\eta} \mathbb{E}_{\pi(A)}\left[\exp\left(\frac{-1}{\lambda}\left(\sum_{t=1}^{H-1} c(s_t, u_t + \epsilon_t) + \lambda \frac{\pi(U + \mathcal{E})}{\bar{\pi}(U + \mathcal{E})} + Q^{\pi^*}(s_H, u_H + \epsilon_H)\right)\right)\right]$$
$$= \frac{1}{\eta} \mathbb{E}_{\pi(A)}\left[\exp\left(\frac{-1}{\lambda}\left(\sum_{t=1}^{H-1} c(s_t, u_t + \epsilon_t) + \lambda \sum_{t=0}^{H} \frac{1}{2}u_t^T \Sigma^{-1}(u_t + 2\epsilon_t) + Q^{\pi^*}(s_H, u_H + \epsilon_H)\right)\right)\right] \tag{19}$$

Note that $\eta$ is the optimal H-step free energy derived in Eq. (10) and can be estimated from $N$ Monte-Carlo samples as

$$\eta = \sum_{n=1}^{N} \exp\left(\frac{-1}{\lambda}\left(\sum_{t=1}^{H-1} c(s_t, u_t + \epsilon_t^n) + \lambda \sum_{t=0}^{H} \frac{1}{2}u_t^T \Sigma^{-1}(u_t + 2\epsilon_t^n) + Q^{\pi^*}(s_H, u_H + \epsilon_H^n)\right)\right) \tag{20}$$

We can form the following iterative update rule where at every iteration $i$ the sampled control sequence is updated according to

$$u_t^{i+1} = u_t^i + \alpha \sum_{n=1}^{N} w(\mathcal{E}_n) \epsilon_n \tag{21}$$

where $\alpha$ is a step-size parameter as proposed by Wagener et al. (2019). This gives us the infinite horizon MPPI update rule. For $H = 1$, this corresponds soft Q-learning where stochastic optimization is performed to solve for the optimal action online. Now we develop soft Q-learning algorithm that utilizes infinite horizon MPPI to generate actions as well as Q-targets.

### 4.3 THE INFORMATION THEORETIC MODEL PREDICTIVE Q-LEARNING ALGORITHM

Since we do not have access to $Q^{\pi^*}$, we can not estimate the importance weight in Eq. (19) exactly. Hence, we consider Q functions parameterized by $\theta$ denoted by $Q_\theta(s, a)$. Similar to deep Q-learning algorithms, we maintain an replay buffer (Mnih et al., 2015), and update parameters by stochastic gradient descent on the loss $L = \frac{1}{K} \sum_{i=1}^{K} (y_i - Q_\theta(s_i, a_i))^2$ for a batch of $K$ experience tuples $(s, a, c, s')$ sampled from the buffer where targets $y_i$ are given by

$$y = c(s,a) - \lambda \log \mathbb{E}_{\pi^*(a_1 \ldots a_H)} \left[ \exp \left( \frac{-1}{\lambda} \left( \sum_{t=1}^{H-1} c(s_t, a_t) + \lambda \log \frac{\pi_t^*}{\bar{\pi}_t} + Q_\theta(s_H, a_H) \right) \right) \middle| s_1 = s' \right] \quad (22)$$

Since the value function updates are performed offline we can utilize large amount of computation (Tamar et al., 2017) to calculate $\pi^*(a_1 \ldots a_H)$. In our case it is obtained by performing the infinite horizon MPPI update in Eq. (21) for multiple iterations starting from state $s'$. This allows for directed exploration at a state which leads to better approximation of the free energy (which is akin to approaches such as Covariance Matrix Adaption, except MPPI does not adapt the covariance). This especially helps in early stages of learning by providing better quality targets than a random Q function. Intuitively, this update rule leverages the biased dynamics model $\hat{P}$ for $H$ steps and a soft Q function at the end learned from interactions with the real system.

At every timestep $t$ during online rollouts, a $H$-horizon sequence of actions is optimized using a single iteration of infinite horizon MPPI update rule in Eq. (21) and the first action is executed on the system. Online optimization with predictive models can lookahead to produce better actions than acting greedily with respect to the biased Q function and makes ad-hoc exploration strategies such as $\epsilon$-greedy unnecessary. Using predictive models for generating value targets and online policy optimization helps accelerate convergence as we demonstrate in our experiments in the next section. Algorithm 1 shows the complete MPQ algorithm.

A closely related approach in literature is (Lowrey et al., 2018) which also uses online MPC and offline value function learning, however they assume access to the true dynamics of the system and do not explore the connection between MPPI and entropy regularized RL and hence do not use free energy targets, even though they use MPPI in their implementation.

---

**Algorithm 1:** MPQ

**Input** : Approximate model $\hat{P}$, initial Q function parameters $\theta_1$, experience buffer $\mathcal{D}$
**Parameter:** Number of episodes $N$, length of episode $T$, planning horizon $H$, number of update
episodes $N_{update}$, minibatch-size $K$, number of minibatches $M$

1 **for** $i = 1 \ldots N$ **do**
2    **for** $t = 1 \ldots T$ **do**
3       $(a_t, \ldots, a_{t+H}) \leftarrow$ Infinite horizon MPPI (Eq. (21))
4       Execute $a_t$ on the real system to obtain $c(s_t, a_t)$ and next state $s_{t+1}$
5       $\mathcal{D} \leftarrow (s_t, a_t, c, s_{t+1})$
6    **if** $i \% N_{update} == 0$ **then**
7       Sample $M$ minibatches of size $K$ from $\mathcal{D}$
8       Generate targets using Eq. (22) and update parameters to $\theta_{i+1}$
9    **return** $\theta_N$ or best $\theta$ on validation.

---

## 5 EXPERIMENTS

We perform experiments to test the efficacy of MPQ in overcoming the shortcomings of stochastic optimal control and model free RL in terms of convergence rate, computational requirements and model bias. We also compare MPQ against domain randomization for learning policies that perform well on systems for which accurate models are not known.

### 5.1 EXPERIMENTAL SETUP

We test our approach on sim-to-sim continuous control tasks based on the Mujoco simulator (Todorov et al., 2012) to study the properties of the algorithm in a controlled manner. The

agent is not provided with the true dynamics parameters, but a uniform distribution over them with a biased mean and added noise. This serves as a reasonable approximation of model bias due to inaccurate measurements of physical quantities. Details of the tasks considered are as follows

1. PENDULUMSWINGUP: the agent tries to swingup and stabilize a pendulum by applying torque on the hinge. The agent is provided with a distribution over the mass and length of the pendulum. The state of the system is given by $(\Theta, \dot{\Theta})$, where $\Theta$ is the angular displacement. The cost function penalizes the deviation from the upright position and angular velocity. The initial state of the system is randomized after every episode which is 10 seconds long.

2. BALLINCUPSPARSE: a sparse version of the ball in cup task inspired from Tassa et al. (2018). Given a cup and spherical ball attached by a tendon, the goal is to swing and catch the ball. The agent can actuate motors on the two slide joints on the cup and is provided with a biased distribution over the mass of the ball, its moment of inertia and stiffness of the tendon. A cost of 1 is incurred at every timestep and 0 if the ball is in the cup. The position of the ball is randomized after every episode which is 4 seconds long. An episode is successful if agent catches the ball in the cup .

3. FETCHPUSHBLOCK: proposed by Plappert et al. (2018), the agent position controls the end-effector of a simulated Fetch robot to push a block to a goal location on the table. The cost is the distance between the center of mass of the block and the goal. We provide the agent a biased distribution over the mass, moment of inertia inertia, friction coefficients and size of the object. An episode is considered successful if the agent gets the block within 5cm of the goal in 4 seconds. The positions of both block and goal is randomized after every episode.

4. FRANKADRAWEROPEN: the agent velocity controls a 7DOF Franka Panda arm to open a drawer on a cabinet. A simple cost function based on Euclidean distance and relative orientation of the end effector with respect to the handle and the displacement of the slide joint on the drawer is used. The agent is provided a biased distribution over damping and frictionloss of robot and drawer joints. Every episode is 4 seconds long after which the agent's starting configuration is randomized. Success corresponds to opening the drawer within 1cm of target displacement.

The parameters we selected to randomize are reasonable in real world scenarios since estimating quantities like moment inertia and friction coefficients is especially error prone. All our experiments are performed on a desktop with 12 Intel Core i7-3930K @ 3.20GHz CPUs and 32 GB RAM with only few hours of CPU training. Q-functions are parameterized with feed-forward neural networks that take as input an observation vector and action. Refer to A.1 for detailed explanation of tasks.

| Environment | True parameters | Biased distribution |
|---|---|---|
| PENDULUMSWINGUP | $m = 1\text{kg}$ 
 $l = 1\text{m}$ | $m = [0.9, 1.5]$ 
 $l = [0.9, 1.5]$ |
| BALLINCUPSPARSE | $m = 0.058\text{kg}$ 
 $I_{xyz} = 1.47 \times 10^{-5}$ 
 $T = 0.05$ | $m = [0.0087, 0.87]$ 
 $I_{xyz} = [0.22, 22] \times 10^{-5}$ 
 $T = [0.00375, 1.5]$ |
| FETCHPUSHBLOCK | $m = 2\text{kg}$ 
 $I_{xyz} = 8.33e - 4$ 
 $\mu = [1, 0.005, 10^{-4}]$ 
 $l = 0.025m$ | bias $= 0.35$ 
 $\sigma = 0.45$ |
| FRANKADRAWEROPEN | frictionloss = 0.1 
 damping=0.1 | bias $= 0.1$ 
 $\sigma = 10.0$ |

Table 1: Details of environment parameters and dynamics randomization. The last column denotes the range for the uniform distribution. $I_{xyz}$ implies that moment of inertia is the same along all three axes. $T$ is the tendon stiffness. For FETCHPUSHBLOCK, the block is assumed to be a cube with sides of length $l$. FETCHPUSHBLOCK and FRANKADRAWEROPEN use uniform distribution for every parameter defined by: mean = bias × true value and range = $[-\sigma \times \text{true value}, \sigma \times \text{true value}]$

## 5.2 BASELINES

We compare MPQ with a fixed horizon $H$ against three baselines: MPPI using same horizon as MPQ and no terminal value function, MPPI using a longer horizon and SOFTQLEARNING. For

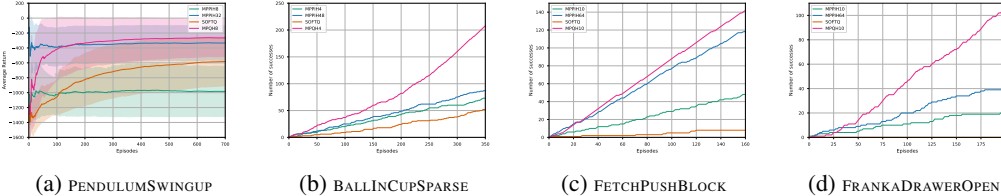

(a) PENDULUMSWINGUP  (b) BALLINCUPSPARSE  (c) FETCHPUSHBLOCK  (d) FRANKADRAWEROPEN

Figure 1: Comparison of MPQ against baselines during training. Baselines are MPPI and SOFTQLEARNING. The number following H in the legend corresponds to horizon for MPC optimization. SOFTQ is equivalent to MPQ with $H = 1$ (a) In the PENDULUMSWINGUP tasks MPQ with $H = 8$ is able to outperform MPPI with $H = 32$ due to the Q function adding global information and correcting for model bias. (b) Due to the sparse nature of BALLINCUPSPARSE task, both SOFTQLEARNING and MPPI with long horizon of $H = 48$ are unable to succeed consistently, whereas the learner is able to outperform them after only a few episodes of training. (c) In the FETCHPUSHBLOCK task, MPQ with $H = 10$ is able to out-perform MPPI using $H = 64$ in merely few minutes of real system interaction. (d) Similarly in FRANKADRAWEROPEN, MPQ with $H = 10$, considerably outperforms MPPI with H=10 and H=64 within a few episodes of training demonstrating scalability to high-dimensional problems. MPQ beats SOFTQLEARNING baseline in all tasks.

SOFTQLEARNING we additionally use a target network to stabilize learning whereas MPQ does not use target networks.

## 5.3 ANALYSIS OF OVERALL PERFORMANCE

### 5.3.1 COMPARISON OF MPQ WITH MPPI AND *soft* Q LEARNING

We test the hypotheses that (1) using a soft Q function as the terminal cost can improve MPPI performance even with a shorter horizon (2) learning from real-data can mitigate effects of model error by adapting to true system dynamics and (3) finite horizon optimization leads to faster convergence as compared to soft Q Learning especially in *sparse* reward tasks. Fig. 1 shows the training curves for MPQ versus *soft* Q learning. Online optimization is able to improve upon the inaccuracies of the Q function and lead to faster convergence. In sparse reward task such as BALLINCUPSPARSE and high-dimensional FETCHPUSHBLOCK and FRANKADRAWEROPEN, SOFTQLEARNING is unable to learn a consistent policy whereas MPQ improves very rapidly. Additionally, an MPQ agent with a short horizon consistently outperforms MPPI with much longer horizon in all tasks. This can be attributed to (1) larger model bias in longer horizon optimization (2) hardness of optimizing longer sequences and (3) global information encapsulated in the Q function. Using a shorter horizon has added computational benefits as well. Since the Q function is learned using data generated from the true system parameters, it is not affected by model bias. In FETCHPUSHBLOCK, the agent outperforms MPPI with H=64 within the first 30 episodes of training which corresponds to roughly 2 minutes of experience on simulation with true parameters. In contrast, SOFTQLEARNING barely ever moves the block and MPPI with H=10 succeeds only when arm is close to initial position of block. Similarly, in FRANKADRAWEROPEN, the agent with $H = 10$ achieves a success rate of more than 5 times as compared to MPPI with H=10 and outperforms MPPI with H=64 as well. The consistent results across all different tasks demonstrates the robustness and scalability of MPQ.

### 5.3.2 BENEFIT OF MPQ OVER DOMAIN RANDOMIZATION

Domain randomization(DR) techniques aim to make the policy learned in simulation robust to modelling errors by randomizing the simulation parameters using manually chosen distributions. However, such policies can be far from optimal on the true system parameters as the learned Q function is inherently biased. However, A Q function learned using rollouts from the real system can overcome model bias. We validate this hypothesis on the BALLINCUPSPARSE task by taking a DR approach inspired by Peng et al. (2018). Simulated rollouts are generated by sampling different parameters at every timestep from a broad range of dynamics parameters in Table 1, whereas real system rollouts use the true parameters. The average success rate reported in Table 2 demonstrates

| Agent | Avg. success rate |
|-------|-------------------|
| MPQH4REAL | 0.85 |
| MPQH4DR | 0.41 |
| MPQH1REAL | 0.09 |
| MPQH4DR | 0.06 |

Table 2: Performance comparison between training using real system rollouts (names ending with REAL) and DR (names ending with DR) on BALLINCUPSPARSE task in terms of average number of successful attempts. Training episodes = 350, test episodes = 100. H is horizon of MPC optimization in both training and testing, where H = 1 is *soft* Q learning.

that a Q function learned solely using DR is unable to generalize to the true system parameters and MPQ has more than twice the success rate when learned on real system.

## 6 DISCUSSION

In this work we have presented a theoretical connection between information theoretic MPC and *soft* Q learning approaches that naturally provides an algorithm to combine stochastic optimal control and model-free reinforcement learning. The theoretical insight not only ties together the different fields, but opens avenues to designing pragmatic RL algorithms that leverage the benefits of both. However, some important questions are yet to be answered. The optimal horizon for MPC is inextricably tied with the model error and optimization artifacts. Investigating this dependence in a principled manner is important for real-world applications. Another interesting avenue of research is characterizing the performance of a parameterized Q function and using it to adapt the horizon of MPC rollouts for smarter exploration.

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

## A    APPENDIX

### A.1    FURTHER EXPERIMENTAL DETAILS

The learned Q function takes as input the current action and a observation vector per task:

1. PENDULUMSWINGUP: $\left[\cos(\Theta), \sin(\Theta), \dot{\Theta}\right]$ (3 dim)

2. BALLINCUPSPARSE: $[x_{ball}, x_{target}, \dot{x_b}all, \dot{x_t}arget, x_{target} - x_{ball}, \cos(\Theta), \sin(\Theta)]$ (12 dim) where $\Theta$ is angle of line joining ball and target.

3. FETCHPUSHBLOCK: $[x_{gripper}, x_{obj}, x_{obj} - x_{grip}, \text{gripper opening}, rot_{obj}, \dot{x}_{obj}, \omega_{obj}$ gripper opening vel, $\dot{x}_{gripper}, d(gripper, obj), x_{goal} - x_{obj}, d(goal, obj), x_{goal}]$ (33 dim)

4. FRANKADRAWEROPEN: $[x_{ee}, x_h, x_h - x_{ee}, \dot{x}_{ee}, \dot{x}_h, \text{quat}_{ee}, \text{quat}_h, \text{drawer}_{disp}$ $d(ee, h), d^{quat}(ee, h), d^{ang}_{ee,h}]$ (39 dim)

For all our experiments we parameterize Q functions with feedforward neural networks with two layers containing 100 units each and `tanh` activation. We use Adam (Kingma & Ba, 2014) optimization with a learning rate of 0.001. For generating value function targets in Eq. (22), we use 3 iterations of MPPI optimization except FRANKADRAWEROPEN where we use 1. The MPPI parameters used are listed in Table 3.

| Environment | Cost function | Samples | $\Sigma$ | $\lambda$ | $\alpha$ | $\gamma$ |
|---|---|---|---|---|---|---|
| PENDULUMSWINGUP | $\Theta^2 + 0.1\dot{\Theta}^2$ | 24 | 4.0 | 0.15 | 0.5 | 0.9 |
| BALLINCUPSPARSE | 0 if ball in cup
1 else | 36 | 4.0 | 0.15 | 0.55 | 0.9 |
| FETCHPUSHBLOCK | $d_{block,goal}$ | 36 | 3.0 | 0.01 | 0.5 | 0.9 |
| FRANKADRAWEROPEN | $d_{ee,h} + 0.08d^{ang}_{ee,h}$
$-1.0 + d_{drawer}/d_{max}$ | 36 | 4.0 | 0.05 | 0.55 | 0.9 |

Table 3: Cost function and MPPI parameters

