# OpenReview forum: "Information Theoretic Model Predictive Q-Learning"
_ICLR.cc/2020/Conference — Reject_

### Official Review · AnonReviewer4 · 2019-10-20
**Official Blind Review #4**

**Rating:** 6

**Review:**

This paper studies how a known, imperfect dynamics can be used to accelerate policy search. The main contribution of this paper is a model-based control algorithm that uses n-step lookahead planning and estimates the value of the last state with the prediction from a learned, soft Q value. The n-step planning uses the imperfect dynamics model, which can be cheaply queried offline. A second contribution of the paper is an efficient, iterative procedure for optimizing the n-step actions w.r.t. the imperfect model. The proposed algorithm is evaluated on three continuous control tasks. The proposed algorithm outperforms the two baselines on one of the three tasks. Perhaps the most impressive aspect of the paper is that the experiments only require a few minutes of "real-world" interaction to solve the tasks.

I am leaning towards rejecting this paper. My main reservation is that the empirical results are not very compelling. In Figure 1, it seems like the proposed method (MPQ) only beats that MPPI baseline in BallInCupSparse. The paper seems remiss to not include comparisons to any recent MBRL algorithms (e.g., [Chua 2018, Kurutach 2018, Janner 2019]). Moreover, the tasks considered are all quite simple. Finally, for a paper claiming to "pave the way for deploying reinforcement learning algorithms on real-robots," it seems important to show results on real robots, or at least on a very close approximation thereof. Instead, since the experiments are done in simulation and the true model is known exactly, the paper constructs an approximate model that is a noisy version of the true model.

I do think that this paper is tackling an important problem, and am excited to see work in this area. I would consider increasing my review if the paper were revised to include a comparison to a state-of-the-art MBRL method, if it included experiments on more complex task, and if the proposed method were shown to consistently outperform most baselines on most tasks.

Minor comments:
* The derivation of the update rule in Section 4.2 was unclear to me. In Equation 17, why can the RHS not be computed directly? In Equation 21, where did this iterative procedure come from?
* "Reinforcement Learning" -- No need to capitalize
* "sim-to-real approaches … suffer from optimization challenges such as local minima and hand designed distributions." -- Can you provide a citation for the "local minima" claim? The part about "hand designed distributions" seems to have a grammar error.
* "Model Predictive Control" -- No need to capitalize.
* "Computer Go" -- How is this different from the board game Go?
* "Entropy-regularized" -- Missing a period before this.
* Eq 1 -- The "\forall s_0" suggests that this equation depends on s_0. Can you clarify this dependency?
* "stochastic gradient descent" -- I believe that [Williams 1992] is a relevant citation here.

----------------------UPDATE AFTER AUTHOR RESPONSE------------------------
Thanks for the detailed response!  The new experiments seem strong, and help convince me that the method isn't restricted to simple tasks. I'm inclined to agree that it's not overly onerous to assume access to an imperfect model of the world. Because of this, I will increase my vote to "weak accept."

In the next version, I hope that the authors (1) include multiple random seeds for all experiments, and (2) student the degree to which model misspecification degrades performance (i.e., if you use the wrong model, how much worse do you do?).

**Experience Assessment:**

I have read many papers in this area.

**Review Assessment: Checking Correctness Of Derivations And Theory:**

I assessed the sensibility of the derivations and theory.

**Review Assessment: Checking Correctness Of Experiments:**

I assessed the sensibility of the experiments.

**Review Assessment: Thoroughness In Paper Reading:**

I read the paper thoroughly.

---

> ### Author Response · Authors · 2019-11-13
> **Response to review (1/2)**
>
> We would like to thank the reviewer for the valuable feedback. We agree with the comment that this is an interesting area of research and believe it requires further attention from the RL and controls community. We address your specific concerns below
>
> *************************
> Experiments
> *************************
>
> “In Figure 1, it seems like the proposed method (MPQ) only beats that MPPI baseline in BallInCupSparse.”
>
> We would like to clear up the confusion regarding the performance of MPQ with respect to baselines. Figure 1 shows the comparison among MPQ, MPPI and Soft Q-Learning during training. In all the tasks, MPQ outperforms MPPI when both use the same horizon after only a few episodes (eg. In FetchPushBlock MPPIH10 versus MPQH10). The results also show that MPQ ultimately performs better than MPPI, even when MPPI uses a much longer horizon. In the Pendulum task, MPQ with horizon (H) = 8 achieves an average reward higher than MPPI with H=32 at the end of the training. In FetchPushBlock, MPQ with H=10 obtains a success rate higher than MPPI with H=64. This demonstrates that MPQ allows us to significantly reduce the MPC horizon which is the major computational bottleneck in MPC and needs to be highly tuned in practice. While a longer horizon generally helps in MPC, optimizing for longer action sequences can be harder and model errors can compound over timesteps. MPQ also outperforms the soft Q learning baseline in all tasks.
>
> We observe similar results in the new FrankaDrawerOpen task, where MPQ with H=10 outperforms MPPI with H=64 after only a few episodes of training demonstrating that the approach is scalable to high-dimensional tasks as well. Please refer to overall comments and paper revision for details of the task.
>
> We believe that the consistent results across all different tasks sufficiently demonstrate the efficacy of the approach.
>
> “Moreover, the tasks considered are all quite simple. “
>
> The three benchmarks, in addition to the Pendulum task, that we used to evaluate our method are standard tasks for real-world manipulation problems that require planning to solve successfully. We believe that these tasks are harder and more realistic proxies for real-world robotics problems than MuJoCO benchmarks such as Ant or HalfCheetah. Please refer to responses to Reviewer 1 and Reviewer 2 for more details.
>
> Comparison against Model-based RL: Learning globally consistent neural network models using approaches such as [Chua 2018, Kurutach 2018, Janner 2019] mentioned by the reviewer is indeed an interesting research problem. The problem is inherently hard to solve primarily because of covariate shift between the training and test distributions i.e the distribution of states visited by the controller at test time can be significantly different from the training distribution [1]. Therefore, in this work we focus on learning a policy that can rapidly adapt to the true system dynamics. Additionally, as shown in [2] even when the true model of the system is available, learning a terminal value function for MPC can significantly improve performance and reduce planning horizon. This implies that our approach is in-fact complementary to Model-based RL and can benefit from more accurate models. However, model learning is beyond the scope of the current work and thus we chose not to compare against the above mentioned algorithms.

---

> > ### Author Response · Authors · 2019-11-13
> > **Response to review (2/2)**
> >
> > ************************************
> > Responses to Minor Comments
> > ************************************
> > “The derivation of the update rule in Section 4.2 was unclear to me. In Equation 17, why can the RHS not be computed directly? In Equation 21, where did this iterative procedure come from?”
> >
> > Eq. 17: In order to directly compute the optimal actions it is required to estimate the expectation with respect to the optimal control distribution defined in Eq. 8. Since this is intractable to sample from directly, we employ importance sampling to convert the optimal action sequence into an expectation over a simpler sampling distribution such as a Gaussian.
> >
> > Eq. 21: The iterative procedure is a result of the fact that we are using finite number of samples to estimate the importance sampling weights. If we had infinite compute, we would not require any iterative procedure and could estimate the optimal controls in one shot.
> >
> > We will make these assertions clear in the final revision.
> >
> > * Eq 1 -- The "\forall s_0" suggests that this equation depends on s_0. Can you clarify this dependency?
> >
> > This implies that the policy should minimize the cost-to-go for every state in the state space i.e it must be globally optimal.
> >
> > We would like to thank the reviewer for pointing out all the typos, places where the language could be improved and relevant citations. We will include these in the revision.
> >
> > References
> > [1] Stephane Ross and J Andrew Bagnell. Agnostic system identification for model-based reinforcement learning. arXiv preprint arXiv:1203.1007, 2012.
> > [2] Kendall Lowrey, Aravind Rajeswaran, Sham Kakade, Emanuel Todorov, and Igor Mordatch. Plan online, learn offline: Efficient learning and exploration via model-based control. arXiv preprint arXiv:1811.01848, 2018\

---

> > > ### Comment · AnonReviewer4 · 2019-11-15
> > > **Reviewer response**
> > >
> > > Thanks for the detailed response! This clears up many of my concerns with the paper. I have two minor experimental questions about your response.
> > >
> > > > " This demonstrates that MPQ allows us to significantly reduce the MPC horizon which is the major computational bottleneck in MPC and needs to be highly tuned in practice"
> > > 1. What are the relative wall clock times of the various methods? If a longer horizon causes substantially larger wall clock time, that would help convince me that avoiding long-horizon MPC is an important aim.
> > >
> > > 2. Can you add error bars to Fig 1b-d (or plot multiple random seeds)?

---

> > > > ### Author Response · Authors · 2019-11-15
> > > > **Author response**
> > > >
> > > > 1. The wall clock times depend on the particular implementation of the MPC algorithm but in the case of MPPI they are generally a function of the number of particles*planning horizon which can prohibit use of larger horizon especially in real-time implementation.  Additional reasons for preferring shorter horizon are the difficulty of optimizing for longer action sequences and compounding model errors.
> > > >
> > > > 2. Fig 1b-d show number of successes which is a binary quantity and hence does not error bars. We will include multiple seed results in final version.

---

### Official Review · AnonReviewer2 · 2019-10-23
**Official Blind Review #2**

**Rating:** 3

**Review:**

This paper builds a connection between information theoretical MPC and entropy-regularized RL and also develops a novel Q learning algorithm (Model Predictive Q-Learning).  Experiments show that the proposed MBQ algorithm outperforms MPPI and soft Q learning in practice.

The paper is well-written. The derivation is clean and easy to understand. Adding a terminal Q function, which makes horizon infinite, is a very natural idea. Most of the derivation is unrelated to the additional Q^pi^* (Eq 11-17) and Eq (18-20) are simply plugging in the formula of pi^*, which is derived at Eq (9), which is also quite natural if one knows the result for finite horizon horizon MPPI (i.e., without the terminal Q). For experiments, I'd like to see some results on more complex environments (e.g., continuous control tasks in OpenAI Gym) and more comparison with recent model-based RL work.

Questions:

1. The experiment setup is that a uniform distribution of dynamics parameters (with biased mean and added noise) are used. Why not using a neural network?
2. Eq (22): The estimation for eta is unbiased, However, 1/eta might be biased, so is the estimator for w(E) also biased?

Minor Comments:
1. Eq (16): should the first term be negated?
2. Page 6, last paragraph: performved -> performed.
3. Page 7, second paragraph: produces -> produce.
4. Algorithm 1, L6: N % N_update -> i % N_update.
5. Appendix A.1, second paragraph: geneerating -> generating.


**Experience Assessment:**

I do not know much about this area.

**Review Assessment: Checking Correctness Of Derivations And Theory:**

I assessed the sensibility of the derivations and theory.

**Review Assessment: Checking Correctness Of Experiments:**

I assessed the sensibility of the experiments.

**Review Assessment: Thoroughness In Paper Reading:**

I read the paper at least twice and used my best judgement in assessing the paper.

---

> ### Author Response · Authors · 2019-11-13
> **Response to review**
>
> Thank you for taking the time to review our paper and providing valuable feedback. We greatly appreciate the fact that you found the paper well-written and the derivation easy to follow. We address your questions and concerns below
> **********
> Experiments
> **********
> “For experiments, I'd like to see some results on more complex environments (e.g., continuous control tasks in OpenAI Gym) and more comparison with recent model-based RL work. “
>
> First, we clarify that all the tasks considered in the paper are continuous control tasks. The FetchPushBlock environment is a part of OpenAI Gym’s MuJoCo robotics environments (github.com/openai/gym/tree/master/gym/envs/robotics). BallInCupSparse is based off of the environment provided by DeepMind Control Suite (github.com/deepmind/dm_control) that also uses MuJoCo.
>
> While designing the experiments our aim was to test the efficacy of MPQ in overcoming the shortcomings of both MPC and model-free Q learning. We considered tasks with sparse rewards and that require long horizon planning. The complexity is further aggravated by biased distribution over dynamics parameters making it hard for MPC to solve. By learning a terminal value function from real data we posit that MPQ should adapt to the true system dynamics as well as truncate the horizon of MPC which is the main computational bottleneck. By using a predictive model for Q learning, we also expect to be able to learn an effective value function with significantly less data as compared to entirely model-free soft Q-learning. Hence, we compare against natural baselines of MPPI and soft Q-learning.
>
> Results on all tasks provide evidence in favor of these hypotheses (Figure 1). In all the tasks, MPQ outperforms MPPI that uses the same horizon after only a few training iterations and ultimately performs better than MPPI using a much longer horizon. The BallInCupSparse experiment demonstrates that MPQ is able to discover the sparse rewards by planning into the future, thus mitigating the need to design a dense reward function. FetchPushBlock requires long planning horizon to figure out a trajectory for the arm that can get the block into the goal location. Using a very simple reward function of distance between block and goal, we observed that MPPI with a short horizon (H=10) is unable to even reach close to the block. MPQ with H=10 is able to achieve a success rate higher than MPPI with a significantly longer horizon (H=64) in only a few minutes of experience gathered on the true system.
>
> We have additionally added another benchmark experiment called FrankaDrawerOpen which is a high-dimensional continuous control problem based on a real-world manipulation task. We observe similar results in this task which further strengthens the arguments in the paper. Please refer to overall comments and paper revision.
>
> We do not compare against model based RL methods that learn neural network models for control as learning globally consistent neural network models adds an additional layer of complexity and is beyond the scope of the current work. We would like to note that MPQ is a complementary approach to model learning and one can benefit from the other.
>
> ********
> Questions
> ********
> Q. The experiment setup is that a uniform distribution of dynamics parameters (with biased mean and added noise) are used. Why not using a neural network?
>
> Physics based simulators such as MuJoCo provide globally consistent dynamics models and are widely used in controls [1] and in sim-to-real approaches [2]. However quantities such as friction coefficients, tensile strength, mass and inertia are hard to estimate accurately. Thus, we chose to randomize them as a reasonable approximation of model bias as often done in dynamics randomization [2,3]. MPQ can be used with neural network dynamics models and learning such models has been the part of significant prior work, but is orthogonal to the problem we are attempting to solve.
>
> Q2. Eq (22): The estimation for eta is unbiased, However, 1/eta might be biased, so is the estimator for w(E) also biased?
>
> Yes, since we use importance sampling, the estimate of the weights will be biased for a finite number of samples, but is asymptotically unbiased.
>
> We would like to thank the reviewer for pointing out the typographical errors. We will correct these in the final revision. We hope that we have sufficiently addressed the reviewer’s concerns and would be happy to answer any further questions.
>
> References
> [1] Tassa, Yuval, Erez, Tom, and Todorov, Emanuel. Synthesis and stabilization of complex behaviors through online trajectory optimization. In Intelligent Robots and Systems (IROS), 2012
> [2] X. B. Peng, M. Andrychowicz, W. Zaremba, and P. Abbeel. Sim-to-real transfer of robotic control with dynamics randomization. CoRR, abs/1710.06537, 2017
> [3] I. Mordatch, K. Lowrey, and E.Todorov. Ensemble-CIO: Full-body dynamic motion planning that transfers to physical humanoids. In IROS, 2015.

---

### Official Review · AnonReviewer1 · 2019-10-31
**Official Blind Review #1**

**Rating:** 3

**Review:**



In this paper, the authors proposed the algorithm to introduce model-free reinforcement learning (RL) to model predictive control~(MPC), which is a representative algorithm in model-based RL, to overcome the finite-horizon issue in the existing MPC. The authors evaluated the algorithm on three environments and demonstrated the outperformance comparing to the model predictive path integral (MPPI) and soft Q-learning.

1, The major issue of this paper is its novelty. The proposed algorithm is a straightforward extension of MPPI [1], which adds a Q-function to the finite accumulated reward to predict the future rewards to infinite horizon. The eventual algorithm ends up like a straightforward combination of MPPI and DQN. The algorithm derivation in Sec. 4 follows almost exact as [1] without appropriate reference.

2, The empirical comparison is quite weak. The algorithm is only tested on three environments and only one (Pendulum) from the MuJoCo benchmark. Without more comprehensive comparison  on other MuJoCo environments, the empirical experiment is not convincing.


Minors:

1, The derivation from 15-16 is wrong. The first term should be negative.

There are many claims without justification. For example:
"Solving the above optimization can be prohibitively expensive and hard to accomplish online."

- This is not true. There are already plenty of work solving the entropy-regularized MDP online [2, 3, 4] and achieving good empirical performance.

"The re-optimization and entropy regularization helps in mitigating model-bias and
inaccuracies with optimization by avoiding overcommitment to the current estimate of the cost."
- There is no evidence that the entropy-regularization will reduce the mode-bias. Actually, as discussed in [4, 5] the entropy regularization will also incur some extra bias.

[1] Grady Williams, Nolan Wagener, Brian Goldfain, Paul Drews, James M Rehg, Byron Boots, and Evangelos A Theodorou. Information theoretic mpc for model-based reinforcement learning. In 2017 IEEE International Conference on Robotics and Automation (ICRA), pp. 1714–1721. IEEE, 2017.
[2] Haarnoja, Tuomas, Aurick Zhou, Pieter Abbeel, and Sergey Levine. "Soft actor-critic: Off-policy maximum entropy deep reinforcement learning with a stochastic actor." arXiv preprint arXiv:1801.01290 (2018).
[3] Nachum, O., Norouzi, M., Xu, K. and Schuurmans, D., 2017. Bridging the gap between value and policy based reinforcement learning. In Advances in Neural Information Processing Systems (pp. 2775-2785).
[4] Dai, Bo, Albert Shaw, Lihong Li, Lin Xiao, Niao He, Zhen Liu, Jianshu Chen, and Le Song. "SBEED: Convergent reinforcement learning with nonlinear function approximation." arXiv preprint arXiv:1712.10285 (2017).
[5] Geist, Matthieu, Bruno Scherrer, and Olivier Pietquin. "A Theory of Regularized Markov Decision Processes." arXiv preprint arXiv:1901.11275 (2019).

**Experience Assessment:**

I have published in this field for several years.

**Review Assessment: Checking Correctness Of Derivations And Theory:**

I assessed the sensibility of the derivations and theory.

**Review Assessment: Checking Correctness Of Experiments:**

I assessed the sensibility of the experiments.

**Review Assessment: Thoroughness In Paper Reading:**

I read the paper at least twice and used my best judgement in assessing the paper.

---

> ### Author Response · Authors · 2019-11-13
> **Response to review(1/2)**
>
> Thank you for investing the time to review our paper and providing valuable feedback. We address your questions and concerns below
>
> ******
> Novelty
> ******
> We are interested in solving the infinite horizon entropy-regularized RL problem. This objective helps in incorporating prior knowledge, maintaining exploratory behavior and mitigating the biased update rule in vanilla Q-learning [1]. Model-free RL methods are difficult to implement on real-world systems because of sample inefficiency and safety concerns. In contrast, model-based approaches like MPC bypass these concerns and consistently work well in the real world [2, 3, 4]. The paper focuses on leveraging MPC to solve the infinite horizon entropy-regularized objective. While several previous works have demonstrated the utility of using a Q function or value function as a terminal cost in the MPC objective [6, 7, 8], our paper is the first to show that the principled way to do this is to use a soft Q function. In particular, previous work such as [6], which combines MPPI and, e.g., a vanilla Q function, is not consistent with either the standard infinite horizon RL problem or the entropy regularized RL problem. The key insight in our work is the important connection between entropy regularized RL and information theoretic MPC and the accompanying derivation that shows how methods like MPPI and soft Q learning should be combined.
>
> A few key differences as compared to MPPI [2] and Q learning:
> MPPI
> In Section 4.1 we provide a more straightforward derivation for the optimal sampling distribution as compared to Williams et. al. 2017 who simply assume the expression for the free energy without any justification or motivation and use Jensen’s inequality to upper bound it by the entropy regularized objective. In contrast, our derivation starts from first principles and uses the original infinite horizon entropy-regularized objective to derive the free-energy and optimal policy making the connection between MPPI and soft RL extremely clear. Since the optimal policy is intractable to sample from, we follow the approach of [2] to find the optimal policy from a class of simpler Gaussian policies as presented in Section 4.2. In the original draft we have cited Williams et. al 2016 instead of Williams et. al 2017 [9] accidentally and we will correct this in the revision.
>
> Q-Learning
> Approaches such as DQN are entirely model-free. However, in several real-world robotics problems efficient simulators are available [eg. MuJoCO] that can be leveraged to obtain better Q-value estimates. Previous work in RL such as [5, 6] has demonstrated that given an accurate model of the system, H-step optimization can improve an inaccurate value function. Similar to [6], MPQ leverages MPC not only as the policy executed on the system, but also as a critic, thus making it “Model-Predictive Q Learning”. The following are key distinguishing features from [4]: (1) Our mathematical connection between information theoretic control and soft RL theoretically justifies using MPPI as the control algorithm and free-energy targets for the Q-function. [4] doesn’t explore this connection and uses hard value target despite using MPPI in the implementation. (2) [4] assumes access to an accurate dynamics model whereas in our experiments we explicitly consider model-bias and demonstrate the robustness of MPQ.
>
> *****************
> Empirical Comparison
> *****************
>
> We compared our approach to several different baselines on three different environments. The purpose of the experiments was to investigate the extent to which learning a soft Q function improves over MPC with potentially biased models on realistic robotics manipulation problems. We believe that three benchmarks are sufficient to support and provide insight into the methods and theory detailed in the paper. However, we have now included an additional benchmark, which further supports our conclusions. Please refer to overall comments and revision for details of the task.
>
> The three benchmarks, in addition to the Pendulum task, that we used to evaluate our method are standard tasks for real-world manipulation problems that require planning to solve successfully. We believe that these tasks are harder and more realistic proxies for real-world robotics problems than MuJoCO benchmarks such as Ant or HalfCheetah. Please see response to reviewer 2 for more details.
>
> The BallInCupSparse and FetchPushBlock environments are in fact continuous control tasks that use MuJoCo. As mentioned in Section 5, the environment for the BallInCupSparse task is provided in the DeepMind Control Suite (github.com/deepmind/dm_control) and the one for FetchPushBlock task is available as part of OpenAI Gym’s robotics environments (github.com/openai/gym/tree/master/gym/envs/robotics). The PendulumSwingup environment however uses equations of motion and is not based on MuJoCo. The FrankaDrawerOpen environment is also based on a real-world manipulation task from [10].

---

> > ### Author Response · Authors · 2019-11-13
> > **Response to review (2/2)**
> >
> >
> > **************
> > Minor Comments
> > **************
> > “"Solving the above optimization can be prohibitively expensive and hard to accomplish online."
> >
> > - This is not true. There are already plenty of work solving the entropy-regularized MDP online [2, 3, 4] and achieving good empirical performance. “
> >
> > Thank you for pointing out the relevant work on entropy-regularized RL. We would first like to clarify what we mean by optimizing the policy online. MPC optimizes a finite-horizon sequence of controls calculate the optimal action at every timestep on the system. This results in a state-feedback policy. Efficient MPC approaches are able to perform the optimization in real-time by using simple policy classes, predictive models, truncated horizon optimization and leveraging parallel hardware such as GPUs [2, 4, 9]. Most model-free RL approaches, such as the ones pointed out by the reviewer, use policy gradient or value based learning algorithms to optimize complex policy classes such as deep neural networks. Optimizing them online would be akin to finding the best neural network weights at the current state. Doing so can be prohibitively expensive given the large number of parameters of the network. Hence, these approaches mostly use an iterative process of collecting data from the system and performing SGD on a loss function offline i.e between iterations of interaction. In general, if we assume access to an accurate dynamics model, we could solve for the optimal policy using dynamic programming techniques. However, such techniques suffer from the curse of dimensionality and often need additional assumptions on the form of the dynamics and cost function. Also, exact models are usually not available for real-world robotics problems like manipulation unlike games such as Atari or Go. Efficiently combining model-free and model based approaches can help mitigate these shortcomings which forms the motivation for the work.
> >
> > “"The re-optimization and entropy regularization helps in mitigating model-bias and
> > inaccuracies with optimization by avoiding overcommitment to the current estimate of the cost."
> > - There is no evidence that the entropy-regularization will reduce the mode-bias. Actually, as discussed in [4, 5] the entropy regularization will also incur some extra bias.”
> >
> > We did not intend to make the claim that entropy-regularization will directly reduce model-bias. We will correct this in the final revision. We intended to say that entropy regularization helps in maintaining exploratory behavior and prevents over-optimizing the current estimate of the cost which is biased due to model-errors, finite horizon and limited number of samples.
> >
> > Finally, we would like to thank the reviewer for pointing out typographical errors. We will correct them in the revision. We hope we have sufficiently addressed the reviewers concerns and would be glad to provide any additional clarifications.
> >
> > References:
> > [1] Roy Fox, Ari Pakman, and Naftali Tishby. Taming the noise in reinforcement learning via soft updates. arXiv preprint arXiv:1512.08562, 2015
> > [2] Nolan Wagener, Ching-An Cheng, Jacob Sacks, Byron Boots. An Online Learning Approach to Model-Predictive Control. Robotics:Science and Systems, 2019
> > [3] Pieter Abbeel, Adam Coates, and Andrew Y. Ng. Autonomous helicopter aerobatics through apprenticeship learning. The International Journal of Robotics Research, 2010.
> > [4] Grady Williams, Nolan Wagener, Brian Goldfain, Paul Drews, James M Rehg, Byron Boots, and Evangelos A Theodorou. Information theoretic mpc for model-based reinforcement learning. In 2017 IEEE International Conference on Robotics and Automation (ICRA)
> > [5] Wen Sun, J Andrew Bagnell, and Byron Boots. Truncated horizon policy search: Combining reinforcement learning & imitation learning. arXiv preprint arXiv:1805.11240, 2018.
> > [6] Kendall Lowrey, Aravind Rajeswaran, Sham Kakade, Emanuel Todorov, and Igor Mordatch. Plan online, learn offline: Efficient learning and exploration via model-based control. arXiv preprint arXiv:1811.01848, 2018
> > [7] M. Zhong, M. abd Johnson, Y. Tassa, T. Erez, and E Todorov. Value function approximation and model predictive control. IEEE ADPRL, 2013.
> > [8] U. Rosolia, F. Borrelli, "Learning model predictive control for iterative tasks. a data-driven control framework", IEEE Transactions on Automatic Control, 2017.
> > [9] Grady Williams, Paul Drews, Brian Goldfain, James M Rehg, and Evangelos A Theodorou. Aggressive driving with model predictive path integral control. In 2016 IEEE International Conference on Robotics and Automation (ICRA).
> > [10] Y. Chebotar, A. Handa, V. Makoviychuk, M. Macklin, J. Issac, N. Ratliff, and D. Fox. Closing the sim-to-real loop: Adapting simulation randomization with real world experience. In 2019 IEEE International Conference on Robotics and Automation (ICRA).

---

### Author Response · Authors · 2019-11-13
**Additional experiment and summary of responses**

We would like to thank all reviewers for providing their valuable feedback on our work. We first provide details of an additional benchmark task that we have added to the revision

FrankaDrawerOpen
The agent velocity controls a 7DOF Franka Panda arm to open a drawer on a cabinet. A simple cost function based on Euclidean distance and relative orientation of the end effector with respect to the handle and the displacement of the slide joint on the drawer is used. The agent is provided a biased distribution over damping and frictionloss of robot and drawer joints. Every episode is 4 seconds long after which the agent's starting configuration is randomized. Success corresponds to opening the drawer within 1cm of target displacement.

This task is based on a real-world manipulation problem from [1] and is a high-dimensional continuous control task based on MuJoCo. In order to solve it successfully, the agent requires long horizon planning in 7DOF to calculate a trajectory for the arm to reach the handle and open the cabinet and is hard to solve for model-free RL or MPPI with a short horizon. We use this task to provide further evidence for MPQ’s ability to overcome the shortcomings of both MPPI and soft Q-learning.

Results:
As shown in Figure 1, MPQ with a horizon H=10 outperforms MPPI with the same horizon and achieves five times the success rate by the end of training. It also performs considerably better than MPPI with H=64. This demonstrates that MPQ allows us to truncate the MPC horizon even in high-dimensional problems. We also observe that soft Q-learning is unable to solve any problem. This can be attributed to the general sample inefficiency of model-free RL which is further aggravated in high-dimensional control tasks.

We now provide a summary of responses to the concerns raised by reviewers
******
Novelty
******
The paper provides a principled way to solve the infinite horizon entropy-regularized RL problem and is the first to our knowledge to derive the important connection between information theoretic MPC and entropy-regularized RL. The resulting algorithm is a natural combination of MPPI and soft Q -earning that can overcome the shortcomings of both. While previous approaches such as [2,3] have attempted to combine MPC with learned value functions ([3] even uses MPPI in their implementation), they do not explore this connection and hence are not solving the correct problem in principle. In addition, our experiments provide empirical evidence that Q-functions learned from interactions with the true system can help mitigate the effects of model-bias and add global information allowing us to use a shorter horizon in practice. Using MPC for calculating Q-value targets helps in faster convergence by providing more stable targets early on. This reflects in our experimental results as MPQ learns a competitive policy in few training iterations and considerably outperforms soft Q-learning. We further test the applicability of MPQ in sim-to-real transfer by comparing it to dynamics randomization.

*********************
Experiments and Baselines
*********************
We considered continuous control tasks that require long horizon planning to solve efficiently. The BallInCupSparse, FetchPushBlock and FrankaDrawerOpen are hard tasks and better represent real-world robotics problems than standard MuJoCo benchmarks like Ant or HalfCheetah.

We compare against three baselines: MPPI using same horizon as MPQ (eg. MPQH10 and MPPIH10 in FetchPushBlock), MPPI using a longer horizon and soft Q-learning. These are natural baselines and in all tasks, including 4DOF FetchPushBlock and 7DOF FranksDrawerOpen and the sparse reward BallInCupSparse, MPQ consistently outperforms them. We consider this as strong evidence to prove its robustness, scalability and sample efficiency.

Reviewer 2 and 4 wished to see comparison against model-based RL methods. However, as elaborated in the individual replies, we believe that learning accurate system models is an interesting but entirely different problem and is not within the scope of the current work. In fact, we would like to note that MPQ is a complementary approach to model learning and one can benefit from the other.

We hope that we have satisfactorily addressed all the reviewers' concerns and they would consider updating their ratings. We are looking forward to more fruitful discussions.

References:
[1] Y. Chebotar, A. Handa, V. Makoviychuk, M. Macklin, J. Issac, N. Ratliff, and D. Fox. Closing the sim-to-real loop: Adapting simulation randomization with real world experience. In ICRA, 2019
[2] Kendall Lowrey, Aravind Rajeswaran, Sham Kakade, Emanuel Todorov, and Igor Mordatch. Plan online, learn offline: Efficient learning and exploration via model-based control.arXiv:1811.01848, 2018
[3] M. Zhong, M. abd Johnson, Y. Tassa, T. Erez, and E Todorov. Value function approximation and model predictive control. IEEE ADPRL, 2013.

---

### Author Response · Authors · 2019-11-14
**Update regarding experimental results**

We would like to bring to the attention of all reviewers and meta-reviewers an update to the experimental results in the current version. We have updated results for the following two tasks:

1. FetchPushBlock
We fixed a bug in our previous implementation and re-trained the MPQ and SoftQ for this task. We added comparison against MPPI with H=64 and observe that MPQ with H=10 is able to outperform it within a few episodes of training. (Refer Figure 1 in revision).

2. FrankaCabinetEnv
We added comparison against MPPI with H=64 and observe that the performance of MPQ with H=10 is still considerably better.

Both these results further support the method presented in the paper. We have updated our responses below to match these new numbers and would be glad to answer any questions that the reviewers may have.

---

### Decision · Program_Chairs · 2019-12-19

**Decision:**

Reject

**Comment:**

The authors develop a novel connection between information theoretic MPC and entropy regularized RL. Using this connection, they develop Q learning algorithm that can work with biased models. They evaluate their proposed algorithm on several control tasks and demonstrate performance over the baseline methods.

Unfortunately, reviewers were not convinced that the technical contribution of this work was sufficient. They felt that this was a fairly straightforward extension of MPPI. Furthermore, I would have expected a comparison to POLO. As the authors note, their approach is more theoretically principled, so it would be nice to see them outperforming POLO as a validation of their framework.

Given the large number of high-quality submissions this year, I recommend rejection at this time.